# The Predictive Value of Radiographs and the Pirani Score for Later Additional Surgery in Ponseti-Treated Idiopathic Clubfeet, an Observational Cohort Study

**DOI:** 10.3390/children9060865

**Published:** 2022-06-10

**Authors:** Sophie Moerman, Nienke Zijlstra-Koenrades, Max Reijman, Dagmar R. J. Kempink, Johannes H. J. M. Bessems, Suzanne de Vos-Jakobs

**Affiliations:** 1Department of Children’s Orthopaedics, University Medical Center Groningen, 9713 GZ Groningen, The Netherlands; 2Rehabilitation Center Vogellanden, 8013 XZ Zwolle, The Netherlands; nienkekoen@hotmail.com; 3Erasmus MC, Sophia Children’s Hospital, 3015 CN Rotterdam, The Netherlands; m.reijman@erasmusmc.nl (M.R.); d.kempink@erasmusmc.nl (D.R.J.K.); j.bessems@erasmusmc.nl (J.H.J.M.B.); s.devos-jakobs@erasmusmc.nl (S.d.V.-J.)

**Keywords:** clubfoot, radiograph, Pirani score, additional surgery

## Abstract

There are few validated predictors of the need for additional surgery in idiopathic clubfeet treated according to the Ponseti method. Our aim was to examine if physical examination (Pirani score) and radiographs at the age of three months (after initial correction of the clubfeet) can predict the future need for additional surgery. In this retrospective cohort study, radiographs of idiopathic clubfeet were made at the age of three months. The Pirani score was determined at the first cast, before tenotomy, and at the age of three months. Follow-up was at least five years. The correlation between the radiograph, Pirani score, and the need for additional surgery was explored with logistic regression analysis. Parent satisfaction was measured with a disease-specific instrument. The study included 72 clubfeet (50 children) treated according to the Ponseti method. Additional surgery was needed on 27 feet (38%). A larger lateral tibiocalcaneal angle (i.e., equinus) and a smaller lateral talocalcaneal angle (i.e., hindfoot varus) at the age of three months were correlated with the need for additional surgery. Higher Pirani scores before tenotomy and at the age of three months also correlated with additional surgery. Parent satisfaction was lower in patients who needed additional surgery. Both the Pirani scores and the lateral radiographs are predictive for future additional surgery.

## 1. Introduction

Ponseti’s method is internationally regarded as the gold standard for treating idiopathic clubfeet [1]. This method involves weekly manipulations and casting, followed by a tenotomy of the Achilles tendon. Correction is maintained with foot abduction orthoses until the child is four years old [1]. After initial Ponseti treatment, up to 67% of children need repeated casting and/or additional surgery due to relapse [2,3]. Most clubfeet require limited surgery, such as lengthening of the Achilles tendon or transposition of the anterior tibial tendon. Other clubfeet require more extensive surgery, such as partial or complete posteromedial release. In order to customize the follow-up protocol, identification of high-risk cases is needed [1].

Various factors have been indicated as predictors for the need for additional surgery after Ponseti treatment, such as poor evertor muscle activity and brace non-compliance [4]. The Pirani score system is an instrument for assessing the severity of the initial deformation of clubfoot via physical examination [5]. The Pirani score is based on clinical findings of midfoot and hindfoot deformity. Pirani scores can predict the amount of initial cast needed to correct the foot, but there is controversy as to whether this score can predict the need for additional surgery in the future [4,6,7,8].

Before the Ponseti method became the standard, preoperative radiographs of most children with clubfeet were made in order to optimize surgical planning [9]. Conflicting evidence exists for the ability of these radiographs to predict relapse in Ponseti-treated clubfeet. Kang and O’Halloran found that the angle between the tibia and the calcaneus, which represents equinus deformity, could predict the need for additional surgery [10,11]. However, Richards et al. did not find a relation between this angle and additional surgery rates [12].

Our aim was to examine if the Pirani score and radiographs at the age of three months (after initial correction of the clubfeet) can predict the future need for additional surgery. Furthermore, we assessed if parent satisfaction was related to the need for additional surgery.

## 2. Materials and Methods

### 2.1. Patients

In this retrospective study, we used prospectively collected data to answer our research questions. The electronic database of our hospital was searched for eligible patients. All patients with clubfeet treated via the Ponseti method at Erasmus MC—Sophia Children’s Hospital between March 2012 and June 2014 were eligible. Exclusion criteria were (1) children older than 3 months at presentation, (2) non-idiopathic clubfeet, (3) no radiographs at the age of three months available, and (4) follow-up of fewer than 5 years. Follow-up ended in April 2020. If patients moved or were treated elsewhere, parents were contacted. J.B. applied all casts and took Dimeglio and Pirani scores before the first cast, and Pirani scores before tenotomy and at the moment of the radiograph (age 3 months). Treatment and follow-up were performed according to Dutch guidelines [13].

### 2.2. Radiographs

Anteroposterior (AP) and lateral radiographs were obtained at the age of three months, following the protocol of Simons [9,14,15]. These radiographs were made as part of the standard treatment protocol. The angles were measured by two orthopedic surgeons independently in order to calculate interobserver reliability: (SM, NZK). The average of the two measurements was used to calculate the relation between the angles and the outcome of the clubfeet. The following angles were measured:

On an AP radiograph (Figure 1a):Talocalcaneal angle (AP talocalcaneal): the angle between the long axes of the talus and calcaneus. This measurement describes the eversion of the calcaneus under the talus. A small value indicates hindfoot varus [9].Talo first metatarsal angle (AP talo 1st MT): the angle between the long axis of the talus and 1st metatarsal. This measurement describes forefoot abduction or adduction [9].

On a lateral radiograph with maximum dorsiflexion (Figure 1b):Lateral tibial calcaneal angle (lat tibiocalcaneal): the angle between longitudinal axis of the tibia and the plantar aspect of the calcaneus. This measurement describes equinus deformity.Lateral talo calcaneal angle (lat talocalcaneal): the angle between the long axis of the talus and the plantar aspect of the calcaneus. Parallel lines indicate inversion between talus and calcaneus, thus hindfoot varus [9,16,17].Lateral talo first metatarsal angle (lat talo 1st MT): the angle between the long axis of the talus and the first metatarsal. This measurement describes the presence of cavus deformity.Lateral calcaneal 1st metatarsal angle (lat calcaneal 1st MT): the angle between the plantar aspect of the calcaneus and the long axis of the first metatarsal. This measurement also describes the presence of cavus deformity.Foot dorsiflexion between the tibia and a radiolucent wooden board in maximum dorsiflexion (lat foot dorsiflexion). This measurement describes equinus.

### 2.3. Need for Additional Surgery

Outcomes were measured based on the protocol used by Richards et al. [12]. An excellent result was defined as no additional surgery. A good result was defined as the need for additional Achilles tendon lengthening. A fair result was defined as the need for one or more of the following surgeries: transfer of the tibialis anterior tendon, release of the posterior capsule, or plantar fascia release. These procedures could be combined with Achilles tendon lengthening. A poor result was defined as a full posteromedial release. 

### 2.4. Parent Satisfaction

Parent satisfaction was measured via a disease-specific instrument (DSI), developed by Roye et al. [18] and translated and validated in Dutch by Wijnen et al. [19] The DSI is a ten-item questionnaire designed to measure satisfaction and functional outcome in patients with clubfeet and their parents (Appendix A). The DSI was sent via mail to all parents of patients in the cohort in August 2019. If no reply was received, parents were contacted or asked to fill out the DSI during an outpatient visit.

### 2.5. Statistical Analysis

Categorical data are presented as the absolute number of subjects in each group, along with the percentages. Normally distributed, continuous data are shown as means with a 95% CI of mean and, in the case of a non-parametric distribution, as medians with the interquartile range (IQR). The interobserver reliability between the 2 raters was calculated by assessing the intraclass correlation coefficient (ICC), using a two-way random-effect model with absolute agreement. An ICC below 0.50 was classified as poor, between 0.50 and 0.75 as moderate, between 0.75 and 0.90 as good, and above 0.90 as excellent [20]. Continuous data were analyzed using an unpaired T-test in the case of a normal distribution, and a Mann–Whitney test in the case of a non-parametric distribution. The difference in angles measured on the radiograph and the Pirani scores between the group with additional surgery (good and fair outcomes) and without additional surgery (excellent outcomes) was measured with univariate logistic regression. Multivariate logistic regression with all univariate predictors with a *p* value > 0.2 was performed. A Bonferroni correction was applied to correct for multiple testing, setting the *p*-value at 0.0055.

## 3. Results

### 3.1. Patients

A total of 76 patients were treated for clubfeet during the study period. Ten of these patients were excluded because they had non-idiopathic clubfeet (two meningomyelocele, two arthrogryposis, one neuromuscular disease, one Kniest syndrome, and four other syndromes). Ten patients were excluded because no radiographs at the age of three months were available. Two children presented in our hospital at an age older than three months. Four children were lost to follow-up at the ages of 6, 13, 21, and 36 months, respectively. This left a total of 50 children, with 72 clubfeet, available for analysis. Characteristics are described in Table 1.

### 3.2. Need for Additional Surgery

Overall, 45 feet (63%) had excellent results (no need for additional surgery), 9 feet (13%) had good results (8 feet had a single additional Achilles tendon lengthening, and 1 foot had two additional Achilles tendon lengthenings during follow-up); 18 feet (25%) had a fair result (9 of these feet had tibialis anterior transfers, 7 had a tibialis anterior transfer in combination with another procedure, 1 underwent a tibia rotation osteotomy, and 1 underwent a posterior capsule release); no clubfeet had a poor result (Appendix A). The average age for additional Achilles tendon lengthening was 3.5 years (95% CI of mean 2.5–4.6). The average age at tibialis anterior transfer was 5.8 years (95% CI of mean 5.1–6.4 years). 

### 3.3. Radiographs

For the angles measured on the radiographs, ICC scores between the two raters were moderate or good except for the AP talo first metatarsal angle (0.09 (95% CI −0.37 to −0.41)) (Table 2). Therefore, this parameter was excluded from further evaluation.

We compared the angles measured on the radiographs between the patients with and without the need for additional surgery (Table 3). The lateral tibiocalcaneal angle was smaller in patients without additional surgery. The lateral talocalcaneal angle was larger in patients without additional surgery. The other values showed no differences between the groups. Pirani scores before tenotomy and at the time of radiograph were lower in patients without the need for additional surgery. The multivariate logistic regression did not yield any significant predictors for the need for additional surgery (Appendix A).

At the age of 3 months, 51 out of 67 clubfeet (76%) were fully corrected (i.e., Pirani score = 0) (Table 4). Fully corrected clubfeet (i.e., Pirani score = 0) had a lower risk of additional surgery (30% vs. 69%). In fully corrected clubfeet, the lateral tibiocalcaneal angle on the lateral maximum dorsiflexed radiograph was smaller (56 vs. 71 degrees, *p* < 0.01), and the talocalcaneal angle was larger (33 vs. 24 degrees, *p* < 0.01). 

### 3.4. Parent Satisfaction

A total of 54 DSI scores were collected (74%). The median DSI score was 93 (IQR 79–97). Median DSI satisfaction was 93 (IQR 73–100), and median DSI function was 93 (IQR87–100). There were no differences between the group that filled in a DSI and the group without DSI in sex or laterality (uni- or bilateral). DSI satisfaction was higher in the excellent outcome group (*p* < 0.05) (Appendix A).

## 4. Discussion

We determined the relation between physical examination (Pirani score), radiographs at the age of three months, and the need for additional surgery in 72 clubfeet treated according to the Ponseti method. Additional surgery was needed for 27 feet (38%). A larger lateral tibiocalcaneal angle (equinus), and a smaller lateral talocalcaneal angle (parallelism, i.e., hindfoot varus) at the age of three months were correlated with additional surgery. A higher Pirani score before tenotomy and at three months was also correlated with additional surgery.

The lateral tibiocalcaneal angle describes the position of the calcaneus in relation to the tibia. Therefore, this is a marker for the length of the Achilles tendon and describes residual equinus deformity. The lateral tibiocalcaneal angle was higher (i.e., more equinus) in patients with additional surgery (67 degrees) compared to patients without additional surgery (55 degrees) *p* < 0.05. Previous studies have also demonstrated that the lateral tibiocalcaneal angle before tenotomy [10,11] and after the boots and bars treatment [16] positively correlate with relapse. Maximum foot dorsiflexion, measured on a lateral radiograph, did not correlate with additional surgery. When a midfoot break is present [10], the angle between the tibia and the radiolucent wooden board under the sole of the foot can be small, while the ‘real’ equinus, measured with the tibiocalcaneal angle, is large. This phenomenon might have occurred in our data.

A smaller lateral talocalcaneal angle (parallelism) is an indication of the presence or persistence of inversion between the talus and calcaneus, and thus, hindfoot varus [17]. In our data, a smaller lateral talocalcaneal angle was associated with additional surgery (27.1 vs. 33.6, *p* < 0.05). In addition, a previous study by Shabtai et al. shows that parallelism measured after boots and bars treatment is associated with a higher relapse rate [16]. Richards et al. found that the lateral talocalcaneal angle was larger (i.e., less parallelism) in children with a good outcome, compared to a fair outcome, in 312 clubfeet between the ages of 18 and 24 months (during boots and bars treatment) [12]. Li showed that parallelism decreased after tenotomy, suggesting that a tenotomy can improve subtalar joint alignment [21].

Interrater reliability on the AP radiograph for AP talo first MT was extremely poor in our study [20]. The angle between the talus and the first metatarsal on the AP view was difficult to measure because of the circular shape of the talus on this view at this age. Interrater reliability for angles measured on the lateral radiograph was good or excellent. This is comparable to the recent literature [22].

The Pirani score before the first cast (age usually < 1 week) is known to predict the amount of initial cast needed, but other authors have stated that it cannot be used to predict a future need for additional surgery [4,6,23]. In our study, the Pirani score taken just before the first cast was not predictive of future surgery. A Pirani score taken before tenotomy (average age 8 weeks) and before radiographs (average age 3 months) was predictive for future surgery. 

When we compare the fully corrected feet at the time of radiograph (Pirani score = 0, *n* = 51) to the not fully corrected feet (Pirani score > 0, *n* = 16), additional surgery rates are higher in not fully corrected feet (69% vs. 30%). Furthermore, lateral talocalcaneal angles were smaller (i.e., parallelism, thus hindfoot varus), and tibiocalcaneal angles were larger (i.e., more equinus) in the not fully corrected clubfeet. We found that radiographs at the age of three months show which feet were initially fully corrected. The definition of a residual deformity is a deformity that underwent primary treatment but was never fully corrected and needs additional treatment [24]. Clubfoot relapse is defined as any feature of the clubfoot reoccurring after initially successful treatment, which needs additional treatment [1]. We believe that part of the relapse and need for additional surgery we describe in this study is actually the consequence of residual clubfeet. More awareness of these residual clubfeet at an early age and early treatment with re-Ponseti casting might have lowered the number of cases in need of additional surgery [25].

We suggest that early identification of residual clubfeet can be carried out with a carefully performed Pirani score at three months or a lateral radiograph in maximum dorsiflexion at three months, along with measurement of the talocalcaneal and tibiocalcaneal angle. Sriharsha found high correlations between radiographs and the Pirani score [26]. The limited numbers included in this study do not allow us to prove that the Pirani score is better than the radiograph at predicting additional surgery, but the multivariate data indeed suggest it; moreover, radiation could be spared. More research should be performed to confirm this statement.

Parent satisfaction was very high (93 (IQR 79–97)) when we compared them to satisfaction rates other authors found (65–83%) [27,28,29]. Parents whose children needed additional surgery had a lower satisfaction score. This is of importance, since additional surgery in Ponseti treatment is not regarded as a failure of treatment. When we observe our data, a ceiling effect might have occurred, i.e., the instrument was not able to discriminate differences in mildly impaired individuals [28]. The DSI was developed in 2001 in a surgically treated cohort of clubfoot patients, which are known to have a lower satisfaction rate, while the current cohort was treated with Ponseti casting, and all surgeries were performed extra-articular [18,28,29].

Strengths of this study include the low number of missing data, the length of follow-up (5 to 8 years), and the measurement of parent satisfaction, along with rates of additional surgeries. A limitation is that three patients did not visit the outpatient clinic. Instead, their parents were contacted via telephone. We admit this could have led to an underestimation of the need for additional surgery, as a child could have dynamic supination that is not noticed by the patient or his parents. In 76% of patients, an initial tenotomy was performed, which is less than 85%, as published by Ponseti [1]. However, only three patients in the group that did not receive a tenotomy needed additional surgery (3/27, 11%). Data on brace compliance were not gathered prospectively, although brace compliance is known to be a large predictor of the need for additional surgery [4]. The length of follow-up is considerable, but the need for additional surgery might occur even after this follow-up, especially since the average age of the tibialis anterior transfer was 5.8 years [3]. Finally, the number of included feet is small (72), and the number of explored risk factors is large (9 factors), dictating a Bonferroni correction. With this adjustment, a significant difference was only found in the lateral tibia calcaneal angle and not in the lateral talocalcaneal angle or Pirani scores.

## 5. Conclusions

A careful physical examination at the age of three months, such as an examination using a Pirani score, is a good method for predicting the need for additional surgery in the future. Lateral radiographs of the foot at the age of three months can also be predictive for additional surgery, probably because they reveal residual deformity much the same as the physical examination. We suggest that a lateral radiograph of the foot might aid when the physical examination is inconclusive, but more research has to be performed. 

## Figures and Tables

**Figure 1 children-09-00865-f001:**
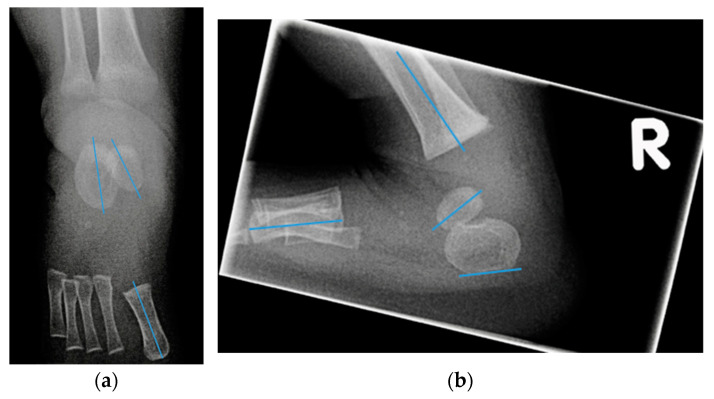
Radiograph obtained at the age of three months: (**a**) anteroposterior radiograph with lines through the longitudinal axis of the talus, calcaneus, and 1st metatarsal; (**b**) lateral radiograph with lines through longitudinal axis of the tibia, the talus, the 1st metatarsal, and the plantar aspect of the calcaneus.

**Table 1 children-09-00865-t001:** Characteristics of the children and feet of children treated according to Ponseti.

	Children (*n* = 50)	Clubfoot (*n* = 72)
Female, *n* (%)	14 (28%)	
Unilateral, *n* (%)	28 (56%)	
Dimeglio score before first cast, median (IQR) ^a^		10 (7–12)
Pirani score before first cast		3.8 (3.4–4.1)
Pirani score before tenotomy ^b^		1.4 (1.1–1.7)
Pirani score before radiograph ^c^		0.3 (0.1–0.5)
Initial tenotomy performed, *n* (%)		55 (76%)
Age at initial tenotomy, weeks		8.3 (7.0–9.5)
Age at radiograph, weeks		14.0 (13.6–14.4)

^a^ 35 feet missing data (Dimeglio registration started September 2013) ^b^ 25 feet missing data, ^c^ 5 feet missing data. Data are presented as means with 95% CI of mean in between parentheses.

**Table 2 children-09-00865-t002:** Angles measured on radiograph and interrater reliability.

	Mean Angle	95% CI of Mean	ICC Average Measure
AP talocalcaneal	16.0	13.3–18.7	0.80 (0.68–0.88)
AP talo 1st MT	9.7	6.8–12.5	0.09 (−0.37–0.41)
Lat tibiocalcaneal	59.1	55.4–62.8	0.99 (0.98–0.99)
Lat talocalcaneal	31.2	28.5–33.9	0.94 (0.90–0.96)
Lat talo 1st MT	−26.9	−31.3–22.5	0.89 (0.80–0.94)
Lat calaneal 1st MT	13.4	11.0–15.7	0.85 (0.76–0.91)
Lat foot dorsiflexion	45.7	41.7–49.7	0.99 (0.99–1.00)

AP = anteriorposterior, Lat = lateral, MT = metatarsal, ICC = intraclass correlation coefficient.

**Table 3 children-09-00865-t003:** Angles measured on radiographs and Pirani scores in patients with and without need for additional surgery.

	No Additional Surgery		Additional Surgery						
	Excellent (*n* = 45)		Good (*n* = 9) Fair (*n* = 18)		*t* Test	Univariate Logistic Regression			
	Mean	SD	Mean	SD	*p*	B	S.E.	Exp (B)	*p*
AP talocalcaneal	16.4	11.0	15.4	12.2	0.75	−0.00	0.02	1.0	0.73
Lat tibiocalcaneal	54.7	12.0	66.5	18.9	<0.05	0.06	0.02	1.06	0.005
Lat talocalcaneal	33.6	11.3	27.1	10.5	<0.05	−0.06	0.03	0.95	0.025
Lat talo 1st MT	−27.3	20.1	−26.1	15.7	0.51	0.00	0.1	1.0	0.79
Lat calaneal 1st MT	13.7	8.4	12.8	11.8	0.75	0.0	0.03	0.99	0.72
Lat foot dorsiflexion	43.5	14.0	49.8	20.0	0.19	0.02	0.02	1.02	0.15
Pirani score before the 1st cast	3.6	1.5	4.1	1.4	0.09	0.30	0.18	1.35	0.099
Pirani score before tenotomy	1.0	0.6	1.9	1.2	<0.05	1.27	0.50	3.56	0.011
Pirani score before radiograph	0.1	0.3	0.7	1.1	<0.05	1.58	0.63	4.83	0.012

**Table 4 children-09-00865-t004:** Comparison between patients with fully corrected clubfeet at the age of three months (Pirani = 0) and not fully corrected clubfeet (Pirani > 0).

	Clubfeet Pirani = 0 *n* = 51	Clubfeet Pirani > 0 *n* = 16	
Lat talocalcaneal (radiograph)	32.9 (11.7)	24.3 (9.2)	*p* < 0.01
Lat tibiocalcaneal (radiograph)	55.8 (12.8)	71.3 (20.5)	*p* < 0.01
Result excellent	36 (71%)	5 (31%)	
Result good	4 (8%)	4 (25%)	
Result fair	11 (22%)	7 (44%)	

## Data Availability

The data presented in this study are available on request from the corresponding author. The data are not publicly available due to privacy.

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
