# Peer review of "The Predictive Value of Radiographs and the Pirani Score for Later Additional Surgery in Ponseti-Treated Idiopathic Clubfeet, an Observational Cohort Study"

_children, 2022, doi:10.3390/children9060865_

Round 1
Reviewer 1 Report
Overall, the authors should expand their literature search as several key studies published over the last 5-8 years are missing. Risk factors related to relapse and the need for anterior tibial tendon transfer surgery using the Ponseti method have been studied and reported in larger cohorts, for example, by several authors. Previous investigators have identified the discussed the relationship between the initial severity of deformity and lack of compliance with bracing as risk factors and this should be discussed and cited.
Detailed comments
Introduction:
Paragraph 2: "At this moment, there are no valid predictors for the need for additional surgery after Ponseti treatment," is not accurate or supported by .current pediatric orthopaedic literature. Please conduct another literature search. (One suggestion- search Zionts, L. et al. He has done a number of studies on this topic over the last decade that are all highly relevant to this manuscript)
Materials and Methods:
Please expand the clinical care regimen. Who applied the casts? How often were patients seen after tenotomy?
Was patient/parent compliance with bracing monitored? Verbal reports? Which brace was used? Compliance with bracing has been well documented to correlated with both relapse and the need for subsequent surgery. If this was not considered or included in this study, this is a major limitation that should be discussed and included in the limitations section.
Was a Dimeglio score used, as well? If not, please discuss, as the Dimeglio score is widely used to assess initial severity.
Results:
If the mean age at TATT was 5.8 years, perhaps a minimum length of follow-up of 5 years is not sufficient to capture need for subsequent surgery. It is likely that a longer duration of follow-up is necessary as some of the patients with less follow-up may still go on to have surgery.
Figure 1: Please clean up this image. It does not appear to have been formatted for publication. There is no figure legend, or explanation of the boxplot.
Reviewer 2 Report
This is a nicely done study looking at predictors for recurrence in idiopathic clubfoot, notably radiographs and Pirani scores. They also assess parent satisfaction. While the cohort was relatively small (50 patients) there is nonetheless value in this study. It is interesting that they found the measurement of AP talo-first metatarsal to be low between the raters but they appropriately excluded this from further evaluation. This was also nicely addressed in the discussion.
I think that the analysis is well done, including a bonferroni correction.
The authors surmise that the Pirani score may be sufficient to predict recurrence without the radiograph, which would be nice to spare the newborns the radiation. Ultimately a prediction score for recurrence of clubfoot could be valuable but is probably outside the scope of this paper given the low numbers.
Round 2
Reviewer 1 Report
I do not feel that the authors adequately addressed the previous concerns raised. They certainly did not submit a "major revision."
In my previous review, I recommended that the authors expand their background and literature search, to say the least, as many authors have identified variables associated with risk of relapsed clubfoot deformity and/or the need for anterior tibial tendon transfer surgery. I believe I even suggested a few names to search. There has been a lot of work on this topic and the authors present the findings from a very small cohort, with limited data collection throughout the treatment period. As such, their findings are of limited value to the community, particularly when they are not placed in proper context with the considerable work that has preceded their study.
Author Response
please see the attachement
